# EFFICIENT VALUE PROPAGATION WITH THE COMPOSITIONAL OPTIMALITY EQUATION

## ABSTRACT

Goal-Conditioned Reinforcement Learning (GCRL) is about learning to reach pre-defined goal states. GCRL in the real world is crucial for adaptive robotics. Existing GCRL methods, however, suffer from low sample efficiency and high cost of collecting real-world data. Here we introduce the Compositional Optimality Equation (COE) for a widely used class of deterministic environments in which the reward is obtained only upon reaching a goal state. COE presents a novel alternative to the standard Bellman Optimality Equation, leading to more sample-efficient update rules. The Bellman update uses the immediate reward and the bootstrapped estimate of the best next state. Our COE-based update rule, however, uses the best composition of two bootstrapped estimates in an arbitrary intermediate subgoal state. In tabular settings, the new update rule guarantees convergence to the optimal value function exponentially faster than the Bellman update! COE can also be used to derive compositional variants of conventional (deep) RL. In particular, our COE-based version of DDPG is more sample-efficient than DDPG in the continuous grid world.

## 1 INTRODUCTION

Reinforcement Learning (RL) in the real world (for robotics etc.) remains difficult (Sünderhauf et al., 2018). A major obstacle is the low sample efficiency of most RL algorithms, compounded by the high cost of data acquisition. Here we focus on goal-conditional RL (GCRL), where a reward is received upon reaching a specified upfront goal (Kaelbling, 1993; Schaul et al., 2015; Schmidhuber, 1990a). We propose a GCRL method to improve sample efficiency in an important class of deterministic environments with a sparse reward received upon reaching a specified upfront goal (Kaelbling, 1993; Schaul et al., 2015; Schmidhuber, 1990a).

| Dist. to the goal | #steps to Opt. Value | | #steps to Opt. Policy | |
|---|---|---|---|---|
| | VI | COE-VI | VI | COE-VI |
| **Empirical results** | | | | |
| 2 | 2 | 1 | 2 | 1 |
| 10 | 10 | 4 | 6.33 | 4 |
| 25 | 25 | 5 | 19.8 | 5 |
| 50 | 50 | **6** | 38.14 | **6** |
| **Theoretical guarantees** | | | | |
| 100 | 100 | 7 | 100 | 7 |
| 1000 | 1000 | 10 | 1000 | 10 |
| 10000 | 10000 | **14** | 10000 | **14** |

Table 1: Learning speed of COE Value Iteration (COE-VI) compared to Value Iteration (VI) on the gridworld environment (See section 6.1).

When goals are distant and widely distributed across the space (Gupta et al., 2019; Hoang et al., 2021; Park et al., 2023), credit for reaching them should be rapidly assigned across long-time lags to relevant, previously executed actions. To achieve this, multi-step methods estimate the value of a policy after unrolling the trajectory for a chosen number of steps. (Precup, 2000; Schulman et al., 2015) These methods, however, have been shown to suffer from high variance (Cortes et al., 2010).

Multi-step methods construct value targets from rewards observed during a rollout and from the estimated value of the state achieved at the end of the rollout. This can be seen as a combination of a Monte-Carlo estimate obtained through the rollout with bootstrapping in estimating the value of the last achieved state. Here, we propose an alternative approach that uses two bootstrapped estimated values–from the current state to a subgoal, and from the subgoal to the goal–to design updates.

To facilitate quick propagation of the value estimates, we revisit the foundation of all RL algorithms - the Bellman Optimality Equation (Sutton & Barto, 2018), and propose an alternative optimality equation. This new equation leverages assumptions about goal-conditioned environments and decomposes a problem into subproblems more efficiently than the Bellman equation. Consequently, its value iteration counterpart exhibits a theoretically provable, exponentially faster convergence rate in tabular environments. We call it the *Compositional Optimality Equation* (COE) because it allows for arbitrary compositions of bootstrapped value estimates. With the COE, we derive methods analogous to widely adapted Deep RL algorithms, DQN and DDPG.

The primary benefit of employing the COE and consequent algorithms is a more efficient value propagation achieved by selecting a more uniform decomposition than the traditional Bellman equation. The Bellman equation divides the value estimates into two parts - a one-step reward $r$ and the bootstrapped estimation of the value gathered throughout the rest of the rollout $V(s')$. This is particularly applicable in scenarios where the final state is unknown or where it is unclear whether a final state even exists. However, to be accurate, it requires the bootstrapped value $V(s')$ also to be accurate. In general, if the distance from the state to the goal is $d$, then accurate bootstrapped estimates of distance $d-1$ are needed for the Bellman estimate to be accurate, leading to linear time propagation of the value estimates.

In this paper, we derive a method tailored for GCRL that enables **exponentially** faster convergence rate in terms of number of updates of the value function for tabular environments. Our method, by imposing hierarchical inductive bias into the value function, is able to utilise long-distance information in the update. As a result, it is able to propagate the values significantly faster than the Bellman update. We extend this result to model-free algorithms with function approximators and empirically demonstrate its convergence in a continuous gridworld environment with stochastic elements.

To summarize, our contributions are:

1. We derive the new *Compositional Optimality Equation* for Goal-Conditioned Reinforcement Learning and prove that it is satisfied by the optimal value function.

2. From the *Compositional Optimality Equation* we derive *Compositional Value iteration* and prove its convergence as well as the fact that the time required to achieve the optimal policy is only logarithmic compared to the linear time of standard Value Iteration. This results in a significantly faster sample efficiency and handling much more distant goals.

3. We generalize *Compositional Value Iteration* to State-Action value function to apply it to model-free scenarios, from which we derive analogous versions of leading algorithms for continuous state spaces with discrete action space (DQN) and continuous action space (DDPG).

4. We empirically demonstrate the sample efficiency benefits of Compositional Value Iteration and Compositional DDPG on gridworld and continuous gridworld environments, respectively.

## 2 RELATED WORK

Deep Reinforcement Learning (RL)(Sutton & Barto, 2018) has emerged as a powerful paradigm for training intelligent agents across a wide variety of applications, ranging from gaming (Mnih et al., 2015; Silver et al., 2017; Vinyals et al., 2019) to natural language processing(Christiano et al., 2017; Ouyang et al., 2022) and robotics (Levine et al., 2016; Popov et al., 2017; Rajeswaran et al., 2017). However, training Reinforcement learning agents continues to be a challenging endeavor due to its substantial energy consumption and intensive computational requirements. These challenges amplify when RL is applied in real-life scenarios, where each sample is considerably more costly compared to simulated samples (Sünderhauf et al., 2018; Pan et al., 2017; Liu et al., 2021).

To alleviate the training costs for real-life robots and RL agents, a series of research papers have proposed strategies to improve sample efficiency. Mai et al. (2022); Dearden et al. (1998) leverage uncertainty to better utilize information from the value functions, while Liu et al. (2023); Faccio et al. (2022); Guo et al. (2021); Arjona-Medina et al. (2019) tackle the credit assignment problem by explicitly identifying crucial states. Wang et al. (2022) propose an alternative to the Bellman Equation, aiming for rapid credit assignment in the context of multi-step off-policy RL. Our method

does not involve multi-step updates; instead, it applies value function compositions. Sample efficiency is also related to the problem of exploration in RL (Amin et al., 2021; Schmidhuber, 1991b; Thrun, 1992; Sutton, 1995)

In goal-reaching tasks (Kaelbling, 1993; Schaul et al., 2015; Schmidhuber, 1990a; Andrychowicz et al., 2017), goals are typically distant and widely distributed. Thus, efficient information propagation is crucial to perform well in such environments—a challenge known as the long-horizons problem (Gupta et al., 2019; Ghosh et al., 2017). Park et al. (2023), like us, addresses the long-horizons problem by creating a flat policy from a single value function. However, it does not introduce a hierarchical inductive bias in the value function to accelerate value propagation.

Our method is closely related to hierarchical RL(Schmidhuber, 1990b; Schmidhuber & Wahnsiedler, 1993; Dayan & Hinton, 1993; Wiering & Schmidhuber, 1997; Schmidhuber, 1991a; Parr & Russell, 1997). Hierarchical RL generates subgoals to aid in problem-solving—after identifying a subgoal, it focuses on reaching the subgoal, thereby forming a hierarchical policy (Dietterich, 2000). In contrast, our method employs a flat policy, utilizing subgoals solely for more efficient information propagation through better problem decomposition. In this context, the closest related work to ours is given by Kaelbling (1993); Dhiman et al. (2018). Our formulation, however, allows us to generalize the algorithm into the deep RL case. Our algorithm can be easily integrated with offline hierarchical methods (Zawalski et al., 2022). A line of work focuses on composing value functions in entropy-regularized reinforcement learning. It has been demonstrated that composing value functions in this setting can approximately solve the intersection of tasks (Haarnoja et al., 2018). Similarly, policies are often hierarchical, offering the benefit of using composed functions to solve new tasks without additional learning (Van Niekerk et al., 2019). In contrast, our approach seeks to utilize the composition of value functions for more efficient training, while maintaining a flat policy structure.

## 3 BACKGROUND

We use the formulation from Sutton & Barto (2018). A discrete-time Markov Decision Process (MDP) is a tuple $\mathcal{M} = (\mathcal{S}, \mathcal{A}, \mathcal{T}, r, \gamma, D)$, where $\mathcal{S}$ denotes the state space, $\mathcal{A}$ represents the action space, $\mathcal{T}(s'|s, a)$ is the density function of the probability distribution for transitioning from state $s$ to state $s'$ after selecting action $a$. The function $r(s)$ is the reward associated with each state (contrary to the most widely used $r(s, a)$), while $\gamma \in (0, 1)$ is the discount factor and $D$ describes the distribution over initial states. A policy is a distribution on the space of actions conditioned on the state, represented by $\pi(a|s)$. The policy interacts with the environment by sampling an action for each given state and applying this action, along with the state, to the transition function.

The value function for a given state $s$ under policy $\pi$ is denoted by $V_\pi(s) = \mathbf{E}[\sum_{i=0}^{\infty} \gamma^t R_{t+i+1}|S_t = s]$, defined as the expected discounted sum of rewards collected with policy $\pi$ from state $s$. Expanding the summation within the expectation for the value function gives us the *Bellman expectation equation* $V_\pi(s) = \mathbf{E}_\pi[R_{t+1} + \gamma v_\pi(S_{t+1})|S_t = s] = \sum_{a \in \mathcal{A}} \pi(a|s) \sum_{s' \in \mathcal{S}} \mathcal{T}(s'|s, a)r(s') + \gamma \mathbf{E}[V_\pi(S_{t+1})|S_t = s]$.

The optimal value function is defined as $V^*(s) = \sup_\pi V_\pi(s)$. The optimal policy $\pi^*$ is any policy $\pi$ that $\forall s \in S \ V_\pi(s) = V^*(s)$. For an optimal value function, we also have the *Bellman optimality equation* $V^*(s) = \max_a \mathbf{E}[R_{t+1} + \gamma V^*(S_{t+1})|S_t = s, A_t = a]$. Analogously, the Q-value function, defined on state-action pairs, is represented as $Q_\pi(s, a) = \mathbf{E}[\sum_{i=0}^{\infty} \gamma^t R_{t+i+1}|S_t = s|S_t = s, A_t = a]$.

In goal-conditioned RL, $\mathcal{S}$ takes on a particular structure, $\mathcal{S}_S \times \mathcal{S}_G$, where $\mathcal{S}_S$ is the original state space, and $\mathcal{S}_G$ is the goal space, where the transition function is assumed to maintain the goal constant throughout the episode. In other words, each transition changes the original state, but the goal is chosen only at the beginning of the episode. Whenever referring to a goal-conditioned state, it will be unpacked into the actual state $s$ and goal $g$. This formulation allows us to easily generalize the MDP to the goal-conditioned scenario by treating the $(s, g)$ pair like an extended state. For example $r(s) : \mathcal{S} \to \mathbf{R}$ becomes $r(s, g) : \mathcal{S}_S \times \mathcal{S}_G \to \mathbf{R}$. In a goal-reaching environment, $r(s, g)$ usually has the form of $r(s, g) = \mathbf{1}_{\{s=g\}}$, or a linear transformation thereof. For convenience, we assume that $r(s, g) = \gamma \mathbf{1}_{\{s=g\}}$. Other definitions, such as $V_\pi(s, g) = \mathbf{E}[\sum_{i=0}^{\infty} \gamma^t R_{t+i+1}|S_t = s, G = g]$ also follow this generalization to (state, goal) pairs. We also assume that the environment terminates after reaching the goal state.

## 4 METHOD

Smart decomposition of large problems into subproblems forms the foundation of efficient algorithms. With this principle in mind, we investigate the classic Bellman Optimality Equation and propose a more efficient, adaptive decomposition method.

The Bellman equation decomposes the value function into two parts: the reward obtained at the immediate next step, and the expected discounted reward gathered over all subsequent steps ($\gamma V(s')$). This intuitive formulation makes learning possible even in scenarios with infinite episodes. However, methods based on the direct application of the Bellman optimality equation necessitate accurate estimates for neighboring states to obtain a reliable estimate for the updated states. Therefore, information propagates linearly in terms of loop iterations or target network updates.

A viable strategy to increase the speed of information propagation is employing multi-step updates that roll out the Bellman equation beyond a single step and use nearer estimates for the update. However, multi-step methods, when applied with off-policy algorithms (Sutton, 1988; Precup, 2000), suffer from high-variance (Cortes et al., 2010). Methods, such as Harutyunyan et al. (2016); Munos et al. (2016), have been proposed to reduce the variance of multi-step methods. We propose an alternative approach that uses the philosophy of multi-step methods but, by bootstrapping, avoids high variance related to the sampled rollout.

In the context of goal-conditioned RL, the agent knows the goal it is trying to reach from the start of the episode. We leverage this information to derive a method that significantly accelerates information propagation, achieved by revisiting the Bellman equation. Our *Compositional Optimality Equation* also partitions the problem into two subproblems that are connected by an interim state that we call a subgoal. Unlike the Bellman optimality equation, our intermediate states (subgoals) do not have to be adjacent to the state being updated. Therefore, they can segment the trajectory into more equidistant parts, thereby enabling exponentially faster value propagation.

### 4.1 COMPOSITIONAL OPTIMALITY EQUATION

We focus on Goal-conditioned RL. We make the assumption that the state space is the same as the goal space, denoted as $\mathcal{S}_S = \mathcal{S}_G$. Recall the extended state space $S = \mathcal{S}_S \times \mathcal{S}_G$. Assume that the underlying MDP is deterministic. In other words, $\mathcal{T}$ is not a distribution, but a function $\mathcal{T} : \mathcal{S} \times \mathcal{A} \to \mathcal{S}$. Let $\Pi_{opt}$ represent the set of all deterministic policies that are optimal, such that $\forall_{s,g \in S, \pi \in \Pi_{opt}} V_\pi(s,g) = V^*(s,g)$. Moreover, $\Pi_{opt}$ is not empty as a global optimal policy always exists. Let $T_\pi(s,g)$ denote a trajectory (a set of visited states) of $\pi$ when run on state $s$ and goal $g$.

A state $subg$ is called a subgoal of $(s,g)$ if there exists a deterministic optimal policy that visits $subg$ on the way from $s$ to $g$. In other words, $subg$ is a subgoal from $s$ to $g$ if it is on some optimal path from $s$ to $g$. Formally:

$$subg \text{ is a subgoal from s to g} \iff subg \in \bigcup_{\pi \in \Pi_{opt}} T_\pi(s,g).$$

We define SUBG(s,g) = $\bigcup_{\pi \in \Pi_{opt}} T_\pi(s,g)$ as the set of subgoals from s to g.

An alternative way to define the same set $SUBG$ involves value functions instead of trajectories. Specifically:

$$SUBG_V(s,g) = \{s' \in \mathcal{S}_S : V^*(s,g) = V^*(s,s') V^*(s',g)\}.$$

**Lemma 1:** Equivalence of definitions:

$$SUBG(s,g) = SUBG_V(s,g).$$

**Proof** First, we demonstrate why, in Goal-conditioned deterministic environments, we can examine distance functions instead of value functions.

Let $d_\pi(s,g)$ be a random variable representing a distance from $s$ to $g$ under policy $\pi$ measured as the number of steps needed to reach $g$ from $s$, and $\infty$ if $g$ will never be reached under $\pi$ from $s$. In other words, if $T_\pi(s,g)$ is a sample rollout of $\pi$ from $s$ to $g$, then $d_\pi(s,g) = |T_\pi(s,g)| - 1$. We also define $d(s,g) := \min_\pi d_\pi(s,g)$ as the optimal distance from $s$ to $g$.

Directly from the definition we have

$$V_\pi(s,g) = \mathbf{E}[\sum_{i=0}^\infty \gamma^t R_{t+i+1}|S_t = s] = \mathbf{E}[\sum_{t=0}^\infty \gamma^t \gamma \mathbf{1}_{\{t=d_\pi(s,g)-1\}}] = \mathbf{E}[\gamma^{d_\pi(s,g)}].$$

Since we operate on deterministic environments and focus on deterministic policies, $V_\pi(s,g) = \gamma^{d_\pi(s,g)}$. Thus, $V^*(s,g) = \max_\pi V_\pi(s,g) = \gamma^{\min_\pi d_\pi(s,g)} = \gamma^{d(s,g)}$.

To show that $SUBG(s,g) = SUBG_V(s,g)$ we must demonstrate:

$$V^*(s,g) = V^*(s,subg)V^*(subg,g) \iff \exists_{\pi \in \Pi_{opt}} subg \in T_\pi(s,g).$$

Starting with the left-to-right implication. Let $\pi$ be a deterministic, optimal policy. Then from the assumption we have that $V_\pi(s,g) = V_\pi(s,subg)V_\pi(subg,g)$, which is equivalent to $d^\pi(s,g) = d^\pi(s,subg) + d^\pi(subg,g)$. Therefore, $subg$ must belong to $T_\pi(s,g)$, as otherwise $d^\pi(s,g) \leq d^\pi(s,subg)$ contradicts the sum of distances. Thus, $\pi$ is the policy from $\Pi_{opt}$ such that $subg \in T_\pi(s,g)$

The proof in the opposite direction is also straightforward. Let $\pi$ be a policy that satisfies the right-hand side assumptions, with $\pi \in \Pi_{opt}$ such that $subg \in T_\pi(s,g)$. Note that $T_\pi(s,subg) \bigcup T_\pi(subg,g)$ does not necessarily have to be equal to $T_\pi(s,g)$. However, because $\pi$ is optimal everywhere, we must have $d_\pi(s,g) = d_\pi(s,subg) + d_\pi(subg,g)$ for $subg$ is in $T_\pi(s,g)$. Otherwise, $\pi$ would not be optimal at $(s,subg)$ or at $(subg,g)$. Therefore, $V^*(s,g) = V_\pi(s,g) = \gamma^{d_\pi(s,subg)+d_\pi(subg,g)} = V_\pi(s,subg)V_\pi(subg,g) = V^*(s,subg)V^*(subg,g)$.

**Triangle inequality**  The decomposition of the value function into the product of values at any state (not necessarily optimal) satisfies the triangle inequality.

$$\forall_{s'} V^*(s,g) \geq V^*(s,s')V^*(s',g). \tag{1}$$

This emerges from the triangle inequality of distances expressed in the exponential form: $V^*(s,g) = \gamma^{d(s,g)} \geq \gamma^{d(s,s')+d(s',g)} = \gamma^{d(s,s')}\gamma^{d(s',g)} = V^*(s,s')V^*(s',g)$.

Consequently, we obtain an alternative optimality equation.

**Compositional Optimality Equation**

$V^*(s,g) = V^*(s,subg)V^*(subg,g)$ for any subg $\in SUBG(s,g) = \max_{subg \in S} V^*(s,subg)V^*(subg,g)$

We base our algorithm for subgoal discovery and an alternative update rule on this equation.

## 4.2  COE-VALUE ITERATION

The new optimality equation enables the definition of a new Compositional update rule for value iteration.

---

**Algorithm 1** Generalized Compositional Value Iteration

---

1: Initialize $V = 0$
2: **for** $s \in S, a \in \mathcal{A}$ **do**
3:     $V(s, \mathcal{T}(s,a)) = \gamma$
4:     $V(s,s) = 1$
5: **end for**
6: **repeat**
7:     **for** $s \in S, g \in S$ **do**
8:         $subg \leftarrow$ ProduceSubgoal
9:         $V_2(s,g) \leftarrow V(s,subg)V(subg,g)$
10:     **end for**
11:     $V \leftarrow V_2$
12: **until** convergence

---

**Value iteration as a special case of Generalized Compositional Value Iteration** If ProduceSubgoal yields $subg \leftarrow \arg\max_{a \in \mathcal{A}} V(s, \mathcal{T}(s, a)) V(\mathcal{T}(s, a), g)$ then it is equivalent to the standard Bellman iteration.

**COE-Value iteration** COE-Value Iteration (COE-VI) is achieved when for ProduceSubgoal we select $subg \leftarrow \arg\max_{subg \in S} V(s, subg) V(subg, g)$. Therefore, for COE-Value Iteration, lines 6 and 7 can be rewritten shortly as $V_2 \leftarrow \max_{subg \in S} V(s, subg) V(subg, g)$.

For COE-VI, it is necessary that the value estimates are initialized with values smaller than the optimal values because $V(s, g)$ can not decrease after an iteration of COE-VI. As a consequence, apart from the values that we a priori know, we initialize the value function estimate with zeros.

### 4.2.1 PROOF OF CONVERGENCE OF COMPOSITIONAL VALUE ITERATION

Let $V^k(s, g)$ be the estimate of $V^*(s, g)$ after k steps. We will show by induction that after $k$ iterations of the algorithm, all pairs of states that are closer to each other than $2^k$ have already reached optimal value function as their $V^k(s, g)$ estimates. We will also show that all estimates are smaller than or equal to the optimal value function. In other words, after $k$ iterations we have two conditions satisfied: 1) $\forall_{(s,g):d(s,g)\leq 2^k} : V^k(s, g) = V^*(s, g)$ and 2) $\forall_{(s,g)} V^k(s, g) \leq V^*(s, g)$.

After 0 iterations (upon the first time visiting line 7), we have $\forall_{(s,g):d(s,g)=1} V^0(s, g) = \gamma$ and $\forall_{(s,g):d(s,g)=0}$, also $\forall_{(s,g):d(s,g)>1} V^0(s, g) = 0 \leq V^*(s, g), \forall_s V^0(s, s) = 1 \leq V^*(s, s)$. Thus, the base of the induction is satisfied.

Now for the $k$'th step, assume that our assumption holds for $k-1$. First, $\forall_{s,g} V^{k-1}(s, g) \leq V^*(s, g)$, so as a consequence of triangle inequality 1 after the update $V$ will also satisfy $V^k(s, g) \leq V^*(s, g)$.

Let us consider $(s, g) : d(s, g) \leq 2^k$. Then, let us choose a subgoal from some optimal trajectory that is in the middle, such that $d(s, subg) \leq 2^{k-1}$ and $d(subg, g) \leq 2^{k-1}$. From our assumption $V^{k-1}(s, subg) = V^*(s, subg), V^{k-1}(subg, g) = V^*(subg, g)$. Therefore if $subg$ is selected for an update, we will have $V^{k-1}(s, subg)V^{k-1}(subg, g) = V^*(s, subg)V^*(subg, g) = V^*(s, g)$. However, as we are maximizing over all possible subgoal states, it follows that $V^k(s, g) \geq V^*(s, g)$, since we have to choose a state that yields a product at least as large as $subg$.

In summary, $V^k(s, g) \geq V^*(s, g)$, but also $V^k(s, g) \leq V^*(s, g)$. So $V^k(s, g) = V^*(s, g)$ for any $(s, g)$, such that $d(s, g) \leq 2^k$ which proves the inductive step.

## 5 GENERALIZATION TO DEEP RL

Analogously to how popular Deep RL algorithms are derived from the Bellman Optimality Equation, it is possible to derive COE-based equivalents. In this section, we describe the process of obtaining deep RL algorithms from our tabular COE Value Iteration algorithm (refer to algorithm 1). We provide examples with pseudocode for COE-DDPG and COE-DQN, detailed further in appendix A.

### 5.1 ACTION-VALUE FUNCTION

Transitioning from a planning approach to a model-free learning algorithm requires generalizing the previously introduced equation. This generalization involves adapting it into a state-action value function update capable of directly utilizing experience samples.

Given that our environment is deterministic, the relationship $Q^\pi(s, a) = r(s') + (1 - d)\gamma V_\pi(s')$ holds, where $s' = \mathcal{T}(s, a)$, and the boolean value $d$ is equal to 1 when the episode ends.

The definition of optimal trajectories can be subtle. In the expression $Q(s, a, g)$, committing to the action $a$ might lead to inconsistencies if the discovered subgoal that maximizes $V(s, subg)V(subg, g)$ does not entail selecting action $a$. Therefore, it is necessary to use optimal trajectories from state $s$ to goal $g$ that select $a$ as the initial action. In deterministic environments, this expression can be simplified with $V(s', subg)$, where $s' = \mathcal{T}(s, a)$. Nonetheless, to ensure our update remains model-free, we condition our subgoal generator function, ProduceSubgoal, on

action $a$ as well. The new declaration becomes ProduceSubgoal: $S \times \mathcal{A} \times S \rightarrow S$:

$$Q(s, a, g) \leftarrow Q(s, a, subg) \max_{a_2} Q(subg, a_2, g) \text{ for } subg \text{ in optimal trajectories from } s' \text{ to } g$$

## 5.2 SUBGOAL GENERATOR

The main difference between Bellman-based algorithms and COE-based algorithms is their bootstrapped estimator. Traditional algorithms following the Bellman style decompose rewards into immediate next steps combined with an estimate of the remaining rewards. Contrarily, COE-based algorithms utilize two bootstrapped estimates of subproblem values, as described above.

A critical aspect of the algorithm is the decomposition of the problem, specifically in selecting an appropriate subgoal. In gridworld scenarios, this selection is guided by the maximization of the product of value estimates. The state that maximizes this product is chosen as the subgoal, and its estimate is subsequently used for the update. However, in continuous state environments, attempting direct maximization is infeasible due to the extensive nature of the space, which could lead to an extreme overestimation bias, as mentioned by Van Hasselt et al. (2016). To tackle this, we introduce a separate neural network aimed at predicting the state where the product of value functions is maximized.

We call this network the Generator, denoted as $G : \mathcal{S_S} \times \mathcal{A} \times \mathcal{S_G} \rightarrow \mathcal{S_S}$, as it generates a subgoal state based on the current state, action and goal. The Generator is parametrized by $\theta_g$. The generator is trained by maximizing the state-action version of the product of values defined in COE 4.1. Therefore, at each step, the generator is updated via gradient ascent according to the following gradient expression (Symbols defined as in A):

$$\nabla_{\theta_g} \frac{1}{|B|} \sum_{(s,a,r,s',g,d) \in B} (1-d)Q(s', a, subg)Q(subg, \pi(subg, g), g) \text{ where } subg = G(s, a, g),$$

where $d$ is the flag signalizing whether $s'$ is terminal.

## 5.3 GROUNDING UPDATE

In the Algorithm 1, the initialization with $V(s, \mathcal{T}(s, a)) = \gamma$ is crucial for the algorithm to work. It actually gives the algorithm information about the environment dynamics as well as works as a basis for the induction from which all combined values between pairs of states and goals are derived in the proof 4.2.1.

To mimic this basis for composition in deep RL, we create the so-called "Grounding Update". We train the value function to equal $\gamma$ for the transitions from the experience replay buffer with the next state as a goal. Therefore, at each step, apart from the standard COE update, we also perform the grounding update by taking a step in the direction of descent with the following gradient:

$$\nabla_{\theta_q} \frac{1}{|B|} \sum_{(s,a,r,s',g,d) \in B} (Q(s, a, s') - \gamma).$$

## 5.4 BELLMAN GENERATOR INITIALIZATION

It is possible to simulate Bellman-style estimates with COE-based methods. For this to occur, the problem must be decomposed into the immediate next step and the remainder, akin to Bellman-based updates. In other words, the selected subgoal state must be equal to the next visited state. More formally, in situations where the generator satisfies $G(s, a, g) \approx \mathcal{T}(s, a)$ and that grounding update is well fitted ($Q(s, a, s') \approx \gamma$), COE-based estimates approximate Bellman-based estimates:

$$Q(s, a, g) = Q(s, a, G(s, a, g)) \max_{a'} Q(G(s, a, g), a', g) \approx Q(s, a, s') \max_{a'} Q(s', a', g) \approx \gamma \max_{a'} Q(s', a', g)$$

where $s' = \mathcal{T}(s, a, g)$.

|  | Mean | Min | Max |
|---|---|---|---|
| COE-DDPG | 19.125 | 11 | 35 |
| DDPG | 32.187 | 24 | 38 |
| Relative gain | **40.6%** | **54.2%** | 7.9% |

Table 2: Comparison of sample efficiency of COE-DDPG with DDPG on the continuous gridworld. The table is a summary of 32 runs per algorithm. For each run, we calculate the first epoch in which all episodes were solved. Relative gain is the percentage difference between COE-DDPG and DDPG. Both average-case and best-case are significantly better for COE-DDPG than DDPG.

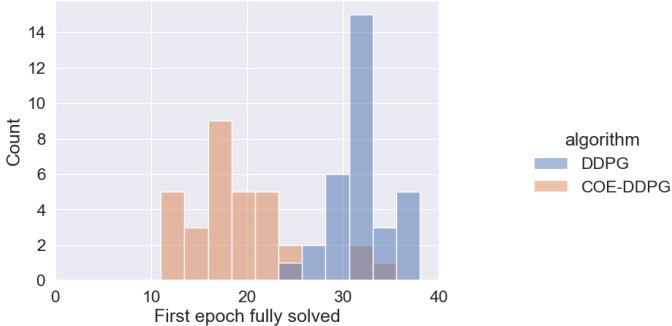

Figure 1: Detailed histogram of results shown in Table 2 comparing sample efficiency of COE-DDPG with DDPG on the continuous gridworld environment. The distribution corresponding to COE-DDPG is shifted to the left in comparison to DDPG. The plot also shows that similar worst-case results are caused by outliers in the COE-DDPG runs.

We exploit this fact to stabilize the training at the beginning when value functions are not reliable enough for subgoal selection. Specifically, we initialize the generator so that the update at the start is the same as the Bellman update. As training progresses, the generator diverges into more distant areas, enabling the utilization of COE's efficiency benefits. With this perspective, the subgoal generator can be viewed as a generalization of the world model (Ha & Schmidhuber, 2018).

## 6 EXPERIMENTS

In this section, we describe the experimental setup used to demonstrate the empirical results of our algorithm, together with the implementation details.[1] We compare the sample efficiency of *COE Value iteration* with *Bellman Value Iteration*, and *COE-DDPG* with DDPG. We evaluate them on gridworld environments - discrete gridworld for value iteration and continuous gridworld for DDPG. Our results in both settings show that our COE-based algorithms are superior to their Bellman counterparts in terms of the number of updates required to solve the environment.

### 6.1 GRIDWORLD

We evaluate our COE Value Iteration (COE-VI) on a two-dimensional gridworld of size $50 \times 50$ and compare it to standard Value Iteration (VI). For each generated board, each position is designated as a wall with a probability of $0.3$, and as a floor tile otherwise. Subsequently, we sample the start state and the goal state uniformly from all floor positions. Next, we check algorithmically whether the goal is reachable from the start state. If not, then we repeat the sampling procedure. If it is, then the RL agent receives it as a task. Our results are consistent with theoretical derivations and confirm that COE Value Iteration is exponentially faster in terms of the number of updates. The results are presented in Table 1. Notice that COE-VI requires fewer updates to solve a problem with a state-to-goal distance of $10000$ than VI requires to solve one of distance $25$. The number of steps

---

[1]For reproducibility purposes, we will publish the code upon acceptance.

necessary to learn to reach the goal at a distance $d$ with COE-VI is exactly $\lceil log_2(d) \rceil$, whereas for standard VI, it is $d$.

## 6.2 CONTINUOUS GRIDWORLD

To evaluate the sample efficiency of Deep RL methods derived from COE, we compare COE-DDPG with DDPG in a continuous gridworld environment. Continuous gridworld also has elements of stochasticity which allow us to validate that despite theoretical results limited to deterministic environments the algorithm can still work in a stochastic setting. The continuous gridworld used for the comparison has a state space of $[-20, 20]^2$, and an action space of $[-1, 1]^2$. In each episode, the start and the goal states are uniformly sampled from the state space. A small centralized Gaussian noise with a standard deviation of 0.1 is added to the action at each step. The goal is considered achieved if the distance between the player's position and the goal position is less than 0.5.

For our agent, we use COE-DDPG (A).We compare the results of COE-DDPG to the standard DDPG (Lillicrap et al., 2015). All models—policy, state-action value functions, and generator—are dense neural networks. Both the policy and generator contain three hidden layers, whereas the value function has four. All models have the same hidden dimension of 512. The generator and policy employ the tanh activation function with suitable scaling. For the value function, the sigmoid activation function is used. The learning rates, denoted as $lr_q, lr_\pi, lr_G$ are consistent and set to $10^{-5}$. All models are trained with the Adam optimizer (Kingma & Ba, 2014).

To evaluate the performance, we execute 32 experiments for each algorithm and study the convergence speed of both. Training starts with 50,000 explorative steps. During this phase, the agent interacts with the environment using a randomly initialized policy without improvement, solely collecting data for the experience replay buffer, which has a capacity of $50,000$. Subsequently, the initial epoch of training begins. This epoch comprises 3000 network update steps and 5000 steps of interaction with the environment, accumulating data for the experience replay (Lin, 1992). Each episode is limited to 50 steps, after which it is truncated. We measure the number of epochs necessary to solve the problem. Specifically, we record the initial epoch that achieves a $100\%$ success rate across its episodes.

The results are shown in Table 2. On average, COE-DDPG proves to be $40.6\%$ faster than its conventional counterpart, with the fastest run being $54.2\%$ faster than the original. In the least favorable scenarios, the advantage is smaller, although closer inspection reveals that these least favorable runs are outliers for COE-DDPG. A more in-depth view of the distribution of runs can be seen in Figure 1. This Figure indicates that the distribution associated with COE-DDPG is distinctly shifted to the left.

While we observe significant improvements from using COE-DDPG over using standard DDPG in this setting, they do not match the exponential speedup witnessed in the discrete case. This disparity results from three factors. First, due to their smoothness, neural networks propagate value updates into nearby states, mitigating the exponential speedup effect. Next, in a discrete environment, the update encompasses all states. In contrast, in a continuous setting, the variance introduced by the generator increases the variance of the entire method. Finally, the discrete algorithm assumes that the values are initialized at $0$. This assumption is problematic in function approximation as it may lead to either vanishing gradients or other complex formulations.

## 7 LIMITATIONS AND FUTURE WORK

Our novel alternative to the Bellman Optimality Equation allows for a more uniform decomposition of values, thereby leading to significantly more sample-efficient algorithms. Our experiments demonstrate a significant speed-up through usage of COE rather than the standard Bellman update. However, in its current form, COE may not be very practical in very complex real-world environments, due to several issues: (1) we assume identical state and goal spaces; (2) our theoretical derivation is limited to deterministic MDPs; (3) there are no results for partially observable environments; (4) the direct operation on states becomes infeasible in higher dimensions. Extending COE's applicability to real-world environments is a promising direction for future research.

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

## A   COE-BASED ALGORITHMS

---

**Algorithm 2** COE-DQN
---
1:  Input: Q-function parameters $\theta_q$, generator parameters $\theta_g$, empty replay buffer $\mathcal{D}$
2:  Set target parameters equal to main parameters $\theta_{q,t} \leftarrow \theta_q, \theta_{g,t} \leftarrow \theta_g$
3:  **repeat**
4:      Observe state $s$ and goal $g$ and select action $a = \max_a Q_{\theta_{q,t}}(s, a, g)$
5:      Execute $a$ in the environment
6:      Observe next state $s'$, reward $r$, goal $g$, and done signal $d$ to indicate whether $s'$ is terminal
7:      Store $(s, a, r, s', g, d)$ in replay buffer $\mathcal{D}$
8:      if $s'$ is terminal, reset environment state.
9:      **if** it's time to update **then**
10:         **if** first time updated **then**
11:             Initialize generator with Bellman Generator Initialization (See 5.4)
12:         **end if**
13:         **for** however many updates **do**
14:             Randomly sample a batch of transitions, $B = \{(s, a, r, s', g, d)\}$ from $\mathcal{D}$
15:             Generate a subgoal for the update $subg = G_{\theta_{g,t}}(s, a)$
16:             Compute targets

$$y(s, a, subg, g) = Q_{\theta_{q,t}}(s, a, subg) \max_{a'} Q_{\theta_{q,t}}(subg, a', g)$$

17:             Update Q-function by one step of gradient descent using

$$\nabla_{\theta_q} \frac{1}{|B|} \sum_{s,a,r,s',g,d) \in B} (Q_{\theta_q}(s, a, g) - y(s, a, subg, g))$$

18:             Update the generator by one step of gradient ascent using

$$\nabla_{\theta_g} \frac{1}{|B|} \sum_{(s,g) \in B} (1-d) Q_{\theta_{q,t}}(s', a, subg) \max_{a'} Q_{\theta_{q,t}}(subg, a', g) \text{ where } subg = G_{\theta_g}(s, a)$$

19:             Update the Q-function by one step of gradient descent on next states from the batch

$$\nabla_{\theta_q} \frac{1}{|B|} \sum_{s,a,s') \in B} (Q_{\theta_q}(s, a, s') - \gamma)$$

20:             Update target networks
21:         **end for**
22:     **end if**
23: **until** convergence

---

---

**Algorithm 3** COE-DDPG

---

1: Input: initial policy parameters $\theta_p$, Q-function parameters $\theta_q$, generator parameters $\theta_g$, empty replay buffer $\mathcal{D}$
2: Set target parameters equal to main parameters $\theta_{p,t} \leftarrow \theta_p, \theta_{q,t} \leftarrow \theta_q, \theta_{g,t} \leftarrow \theta_g$
3: **repeat**
4:     Observe state $s$ and goal $g$ and select action $a = clip(\mu_{\theta_p}(s,g) + \epsilon, a_{Low}, a_{High})$, where $\epsilon \sim \mathcal{N}$
5:     Execute a in the environment
6:     Observe next state $s'$, reward $r$, goal $g$, and done signal $d$ to indicate whether $s'$ is terminal
7:     Store $(s, a, r, s', g, d)$ in replay buffer $\mathcal{D}$
8:     if $s'$ is terminal, reset environment state.
9:     **if** it's time to update **then**
10:        **if** first time updated **then**
11:            Initialize generator with Bellman Generator Initialization (See 5.4)
12:        **end if**
13:        **for** however many updates **do**
14:            Randomly sample a batch of transitions, $B = \{(s, a, r, s', g, d)\}$ from $\mathcal{D}$
15:            Generate a subgoal for the update $subg = G_{\theta_g,t}(s,a)$
16:            Compute targets

$$y(s, a, subg, g) = Q_{\theta_{q,t}}(s, a, subg) Q_{\theta_{q,t}}(subg, \mu_{\theta_{p,t}}(subg, g), g)$$

17:            Update Q-function by one step of gradient descent using

$$\nabla_{\theta_q} \frac{1}{|B|} \sum_{s,a,r,s',g,d) \in B} (Q_{\theta_q}(s, a, g) - y(s, a, subg, g))$$

18:            Update policy by one step of gradient ascent using

$$\nabla_{\theta_p} \frac{1}{|B|} \sum_{(s,g) \in B} Q_{\theta_q}(s, \mu_{\theta_p}, g)$$

19:            Update the generator by one step of gradient ascent using

$$\nabla_{\theta_g} \frac{1}{|B|} \sum_{(s,g) \in B} (1-d)Q_{\theta_{q,t}}(s', a, subg) Q_{\theta_{q,t}}(subg, \mu_{\theta_{p,t}}(subg, g), g) \text{ where } subg = G_{\theta_g}(s,a)$$

20:            Update the Q-function by one step of gradient descent on next states from the batch

$$\nabla_{\theta_q} \frac{1}{|B|} \sum_{s,a,s') \in B} (Q_{\theta_q}(s, a, s') - \gamma)$$

21:            Update target networks
22:        **end for**
23:     **end if**
24: **until** convergence

---

