# OpenReview forum: "Efficient Value Propagation with the Compositional Optimality Equation"
_ICLR.cc/2024/Conference — ICLR 2024 Conference Withdrawn Submission_

### Official Review · Reviewer_7g6n · 2023-10-27

**Soundness:** 3 good
**Presentation:** 2 fair
**Contribution:** 1 poor
**Rating:** 3
**Confidence:** 4

**Summary:**

This paper designs a new update rule for the value function/Q-function for goal-conditioned RL with deterministic environment and binary reward (in other words, the shortest path problem). The update rule exploits the triangle inequality of the distance function by enumerating an intermediate state, as shown below:
$$\forall s',\quad V^*(s,g)\ge V^*(s,s')V^*(s',g).$$ This paper shows that both empirically and theoretically that the number of updates required to find the optimal value function or optimal policy depends logarithmically on the distance between the initial state and the goal state, while standard VI has linear dependency.

**Strengths:**

-	This paper explores an interesting idea of updating the value functions using states other than the next state given by the transition function. And this paper shows that this idea can improve the sample efficiency.

**Weaknesses:**

-	The COE-VI algorithm is heavily restricted to the deterministic shortest path setting.
-	The writing of this paper can be improved. Some of the theoretical claims do not have a formal statement (e.g., the convergence rate of COE-VI, the exponential convergence rate improvement over VI).
-	The claim that COE-VI improves the convergence rate exponentially over standard value iteration could be superficial. If I inferred correctly from Section 4.2.1, the improvement is about the number of rounds that the value function is updated, which is not a standard metric for RL algorithms. In fact, it is unclear to me whether the sample complexity or even the computation complexity of COE-VI is better than vanilla value iteration because each update in COE-VI takes $O(S)$ time where vanilla VI only takes $O(1)$ time (also note that the distance between any two states is upper bounded the number of states in this setting, and vanilla VI converges in $O(S)$ rounds). In addition, the well-known Floyd–Warshall algorithm with doubling trick can already achieve logarithmic updates in the tabular case.
-	The empirically evaluation in this paper is restricted to synthetic toy environments (e.g., 2D-grid). Given that this paper also claims that COE-VI algorithm can be generalized to deep RL, I would expect a more rigorous empirical study on at least a few benchmarking RL environments. In addition, the algorithm in [1] also tries to improve the efficiency of RL algorithms using multi-step updates, which could be a strong empirical baseline for this paper.

[1] He, Frank S., et al. Learning to play in a day: Faster deep reinforcement learning by optimality tightening.

**Questions:**

Please see the weaknesses section.

---

> ### Author Response · Authors · 2023-11-19
>
> We thank the reviewer for their valuable feedback and suggestions. We are glad that the reviewer found our idea interesting.
>
> Answers:
>
> > The COE-VI algorithm is heavily restricted to the deterministic shortest path setting.
>
> We note that the deterministic setting is primarily a consideration for our theoretical analysis (see our general response for further details). We believe that our approach could provide empirical sample-efficiency gains even in stochastic environments. Our experiments in the continuous gridworld show that our approach can handle noisy transitions in practice.
>
> > The claim that COE-VI improves the convergence rate exponentially over standard value iteration could be superficial. If I inferred correctly from Section 4.2.1, the improvement is about the number of rounds that the value function is updated, which is not a standard metric for RL algorithms. In fact, it is unclear to me whether the sample complexity or even the computation complexity of COE-VI is better than vanilla value iteration because each update in COE-VI takes $O(|S|)$ time where vanilla VI only takes $O(1)$ time (also note that the distance between any two states is upper bounded by the number of states in this setting, and vanilla VI converges in $O(S)$ rounds).
>
>  We thank the reviewer for this observation. The COE update indeed requires computing the product for all the states, therefore increasing its computational complexity in the sequential setting by $O(|S|)$. We are interested in the number of updates. One can argue that the update can be easily parallelized, reducing the update cost to $O(\log |S|)$. In practice, however, that could be hard. We focused on the number of updates because our main goal was developing a well-motivated theory for the deep RL case. In this sense, the generator can be seen as an approximation of the subgoal selection from COE-VI, and the generator, of course, has cost $O(1)$.

---

### Official Review · Reviewer_jWih · 2023-10-29

**Soundness:** 4 excellent
**Presentation:** 4 excellent
**Contribution:** 2 fair
**Rating:** 5
**Confidence:** 3

**Summary:**

This paper proposes a value-function like recursion called COE - compositional optimality equation - which extends the goal-conditioned Bellman equation to check over all possible future subgoals instead of just neighboring states. The authors demonstrate that this recursion can find the optimal path to a goal in exponentially fewer steps than standard value iteration, and create an approximation for deep continuous RL where they demonstrate that their method has improved sample efficiency over standard Bellman equation-based algorithms.

**Strengths:**

- The presentation and writing quality is strong. The authors clearly explain the background and theory behind the method, then discuss approximations to large/continuous state spaces with deep RL, and have convincing experiments which show promise in the method for more challenging domains than tabular problems.

- COE shows a significant empirical improvement over standard Bellman-equation based methods.

**Weaknesses:**

- I do not believe the idea of the COE is novel, and there have been previous works which modify the Bellman equation with an inner loop that checks for distances between states i -> j -> k for all states i, j, k. For example, it would be great if the authors could outline the main differences between COE and the work presented in:
"Floyd-Warshall Reinforcement Learning: Learning from Past Experiences to Reach New Goals" Dhiman et. al. 2018.
However, the above paper does not extend their method to large and continuous state spaces, so I still view the contribution of COE-DDPG as novel.

- I also have further questions about the accuracy of the claim that COE converges exponentially faster than standard value iteration. Please see the questions section.

- This paper appears to confuse convergence rate (a type of computational complexity) with sample efficiency / complexity. For example, the paper claims that "the time required to achieve the optimal policy is only logarithmic compared to the linear time of standard Value Iteration. This results in a significantly faster sample efficiency and handling much more distant goals." This is not an implication by any means as these are two completely different quantities.

- While the authors demonstrate their method on continuous state spaces, it is still a gridworld. This paper would be much stronger if the authors could demonstrate their method on more complex problems such as a video game domain or robotic manipulation.

Minor Typos

Introduction: "A major obstacle in training RL agents in real life is the low sample efficiency of the algorithm" -> low sample efficients of RL algorithms?

**Questions:**

"In this paper, we derive a method tailored for GCRL that enables exponentially faster convergence rate in tabular environments. " COE converges faster, but the update is also more expensive to compute as it requires an extra loop over all states to compute the optimal subgoal. So it is as expensive as running value iteration for |S| additional times. Given an equivalent computational budget would this mean that both standard Bellman updates and COE solve the shortest path problem?

---

> ### Author Response · Authors · 2023-11-19
>
> We thank the reviewer for the valuable feedback and suggestions.
> We are glad the reviewer found our paper well-written and our results convincing.
>
> Regarding our contributions, we consider the scalability of our approach to continuous deep RL scenarios as the most important component motivated by our theoretical analysis of COE in deterministic settings.
>
> We thank the reviewer for suggesting the connection to the previously proposed Floyd-Warshall Reinforcement Learning.
> It indeed uses the same decomposition as ours. In future resubmissions, we will revise the paper to clearly highlight our contributions beyond the FWRL and emphasize generalization in deep RL. We have also submitted a new revision addressing the other concerns of reviewers and kindly request feedback on that.
>
> Answers to the question:
>
> > COE converges faster, but the update is also more expensive to compute as it requires an extra loop over all states to compute the optimal subgoal. So it is as expensive as running value iteration for |S| additional times. Given an equivalent computational budget would this mean that both standard Bellman updates and COE solve the shortest path problem?
>
> We thank the reviewer for raising this important issue.
>
> Our main focus was the number of updates, as in the end, in our deep RL implementation, we will use an approximation of the update by using a generator.
> It is true that COE-VI in absolute terms, requires |S| more computations per update. However, some points still make COE favorable.
> These product computations can be very easily parallelized in contrast to sequentially applying Value Iteration.
> Of course, it can be hard to fit the entire state space into the memory.
> This is why we explicitly wrote Generalized Compositional Value Iteration, and not just COE-VI.
> It is up to the algorithm designer to select appropriate candidates for the intermediate state.  Selecting an intermediate state from the set of neighboring states is equivalent to the Bellman update. Selecting it from all possible states gives COE-VI. There are also other choices. For example, one can choose the best state from all states in some neighbors, or have a separate estimator of the best possible state, which gives constant computation time with the possibility of having good interim states at the cost of the additional variance. This is the method we used to generalize to Deep RL.
>
> We will include a discussion related to the additional cost of COE in the updated version of the paper.

---

> > ### Comment · Reviewer_jWih · 2023-11-22
> > **Thank you for the response**
> >
> > Dear Authors,
> >
> > Thank you for the response. I believe with the proposed changes, with a re-focus on scaling this type of method to more challenging continuous/large state spaces (versus gridworld), the paper would be a much stronger future submission.

---

### Official Review · Reviewer_EvmW · 2023-10-30

**Soundness:** 1 poor
**Presentation:** 2 fair
**Contribution:** 1 poor
**Rating:** 3
**Confidence:** 3

**Summary:**

This work proposes a novel update rule for a subset of goal-conditioned MDP problems in the form of a novel optimality equation referred to as Compositional Optimality Equation. The equation incorporates sub-goal information into the update procedure in order to provide an algorithm that is exponentially more efficient than standard Bellman update-based algorithms in finding an optimal value function. The work provides both a theoretical analysis as well as an experimental evaluation in tabular and continuous gridworlds.

**Strengths:**

Motivation
* The work is well-motivated. Sample-efficiency remains a challenge for modern deep RL algorithms and this work tries to find a novel avenue in obtaining policies built on sub-goal discovery.

Structural clarity
* The flow of the paper is good. The theory aims to establish an understanding of the functionality of the method before approximations are introduced and evaluated empirically.

**Weaknesses:**

Contextualization with prior work
The contextualization with respect to prior work is lacking various important areas of contribution and the focus of related work feels off.
* There is a vast literature dating back on value function decomposition that should probably be mentioned in the paper and put into context to strengthen the claim of novelty. Some starting points for classical as well as deep RL works include:

Thomas Dietterich. Hierarchical Reinforcement Learning with the MAXQ Value Function Decomposition. 1999.
Tuomas Haarnoja et al. Composable Deep Reinforcement Learning for Robotic Manipulation. 2018.
Benjamin Van Niekerk et al. Composing Value Functions in Reinforcement Learning. 2019.

* The section of hierarchical methods is rather short and almost solely contains papers from very few authors in a short period of time a long time ago. While the cited work is very valuable, recent work has also made progress on the problem of hierarchical RL. Composition has taken various forms in hierarchical RL and work related to options and skill learning seems more relevant than citations of common RL applications.
* It seems there should be a direct relationship between the presented approach and multi-step methods since the approach is essentially merging multiple steps into a single update. This relationship is not immediately clear.

Mathematical clarity and rigor
* Various definitions are not clearly stated which leads to concerns with the correctness of the provided proof in section 4.2.1. One definition that is quite crucial is that of $V(s, g)$. In section 3, it is stated that the transition function is assumed to maintain a constant goal. So given some state, goal pair $(s, g)$, in the same MDP one can only traverse to some other state with the same goal $(s’, g)$. The value function decomposition now does something like $V(s, subg)V(subg, g)$. This decomposition is a decomposition across two different parts of the MDP where one cannot reach the second part from the first which means these values should be disconnected. So given that only states $g$ are associated with a reward, $V(s, subg)$ must be $0$ and the update is empty which seems to invalidate the reasoning in the proof. In various other places, the second argument to this term seems to change and it is unclear what this means (e.g. line 3 and 4 of the algorithm). More detail on what exactly is happening here would be quite helpful.
* The first part of the induction proof in section 4.2.1 is missing text it seems and also contradicts itself. $V(s, g)$ for all states that are one hop away is supposed to be gamma according to the first definition but for any (s, g) where $s \neq g$, it's supposed to be 0. Clearly, every state one hop away from $g$ is not equal to $g$.
* The conclusion of the proof in section 4.2.1 seems to indicate that at any time the algorithm is optimal which is not true by definition of iteration $0$. Set $k=0$, then $V^k(s, g) = V^0(s, g) = V^∗(s, g)$ is clearly not true.
* The work would benefit from a concisely stated Theorem proposition including a concrete computational complexity term. The claim that the provided algorithm is much faster is not supported well since no runtime complexity analysis is provided. It is also not immediately clear, which part of the algorithm yields the claimed improvement and what details differentiate it from standard value iteration in that sense. The exponential speedup is not characterized properly and it is not clear in which quantity the exponential speedup happens (e.g. exponential in the size of the state space?).
* The work would benefit from a proof or citation to the baseline performance that is claimed to be worse. Given sparse transitions as in the assumption here, the runtime complexity of a single iteration in value iteration is even better than the commonly referenced $O(|S|^2|A|)$ per iteration (see Littman et al.). The number of iterations to convergence is also dependent on $\gamma$ which is not mentioned for either value iteration or the newly presented algorithm. In general, one can rarely find an actual optimal value function $V^*$ but rather stopping criteria need to be used such as $\epsilon$-closeness.  The text does not explain where the theoretical convergence numbers for value iteration come from. For analyses of standard value iteration runtime complexity one can refer to

Littman et al. On the complexity of solving Markov decision problems. 1995.
Nikhil Balaj et al. On the Complexity of Value Iteration. 2019

Strong Assumptions
* The assumptions of the contribution are very strong. As such, it is debatable whether the provided setting should still be considered reinforcement learning. Transitions are assumed deterministic, policies are assumed to be deterministic, the reward functions are very simplified and it seems that only a single goal condition is ever considered (in other words, only a single reward state exists). Fixing all these assumptions, it seems that the problem is closer to a shortest path graph search problem rather than reinforcement learning. Since there is only one reward and no stochasticity, the best thing to do is to find the shortest path from one state $s$ to $g$ on a graph constructed based on S, A and P.

Experimental evaluation
* The experimental evaluation is done on two toy domains that are not very complicated and it is not clear that the approach is scalable to complex domains such as robotics for which the framework is being developed. While the experiments might hint that there is something useful in the formulation, it is not clear how the approach might be scaled up given the strong constraints on its development
* The experiments for a sub-goal based approach would likely benefit from comparisons to other sub-goal approaches from the literature. The literature on subgoal, hierarchical and decomposable RL has focused on speeding up training RL convergence and including baselines from these domains would make a stronger case for the usefulness of the approach.

Textual clarity suggestions
* The notation “subg” is rather lengthy, it might make sense to introduce a symbol for it.
* The assumption that all policies should be deterministic is not stated anywhere explicitly until it is used.
* There are various very lengthy equations that end up as in-line equations. These might be significantly easier to read as non-inline equations.
* Expectations in section 3 should not only be over the policy but also the transitions, otherwise nothing outputs a random variable over $S$.
* The claim that value iteration is a special case of compositional value iteration seems to be overstated since the framework you propose makes significantly stronger assumptions than standard value iteration.
* Page 5, line one, d is not a random variable because everything is deterministic.

Overall, I think this paper is relatively vague and needs to go through another iteration of clarity improvements and contextualization. I am not convinced that the tabular algorithm as presented is correct given various questions around the design and the proof. I will ask for clarification on multiple things in the next section. Finally, the experiments could have stronger baselines. As a result, I’m favoring rejection of the current manuscript.

**Questions:**

Q1. Can you elaborate whether there is a single goal state or not? That seems to be the assumption under the transition dynamics and the first Lemma but becomes unclear later into the manuscript. If only ever one goal is active, shouldn't this problem just be a graph search problem?

Q2. Why is this suggested procedure not just a multi-step update to the Bellman equation? It seems that this should be true with the caveat that the step number differs by state. What differentiates it here?

Q3. Section 5.3 states that a solution to the COE equation is $0$. However, on page 6 the text says that values cannot decrease. Given that values can easily be initialized to be greater than $0$, why would it still be possible to approach the $0$ fixpoint? This indicates to me that something must be off here?

Q4. It also seems that since the update equation is multiplicative and all values are initialized to some value $v \leq 1$, a value can never be larger than $1$. However, if I imagine an MDP with a single state absorbing state that is also the goal, the reward in that state would be $\gamma$ by definition. The maximum value I can obtain can clearly be larger than $1$. Can you explain where my logic is flawed?

Q5. In fact, unless I am missing something, values will always approach $\gamma$ because the largest value product will always be $1 * \gamma$. Why is this not the case?

Q6. In section 5.4, why would Q(s, a, s’) always be equal to $\gamma$? Same equation, I am not sure what “Q_{Bellman}” means? The Q-function should be defined according to section 3.

Q7. If value iteration is a special case of the algorithm you propose, how can your algorithm be significantly faster? Wouldn’t you have to adopt the value iteration runtime in the worst case?

Q8. In the second part of the proof, it states something like “From our assumption V_{k−1}(s, subg) = V^∗(s, subg), V_{k−1}(subg, g) = V^∗(subg, g).” Why can we assume that these are optimal? Nothing prior to that seems to indicate that.

Q9. The update equation in line 9 of the algorithm is missing what is being updated. Am I correct in that $V(s, g)$ is being updated?

---

> ### Author Response · Authors · 2023-11-19
>
> We thank the reviewer for the detailed review, suggestions for improving textual clarity, and references to important related work.
> The feedback helps us improve the paper and understand a reader's point of view.
>
> Answers to the questions:
> > Q1. Can you elaborate whether there is a single goal state or not?
>
> Every episode has a single goal, but the goals can differ across episodes. We assume that the transition function doesn’t change the goal to treat the (state, goal) pair as an extended state and utilize the previously introduced definitions. Therefore, it is possible to view V(s,g) = V(s, subg)V(subg, g) as the decomposition of one episode into two subepisodes and using the combined reward.
>
> > Q2. Why is this suggested procedure not just a multi-step update to the Bellman equation? It seems that this should be true with the caveat that the step number differs by state. What differentiates it here?
>
> This is an important question, and we thank the reviewer for raising this point. Multi-step updates also aim to have more equidistant updates in relation to the state and the goal. However, they rely on Monte-Carlo rollouts, and as a consequence, they are prone to high variance, as shown in past work [1,2,3]. Here, we propose an alternative approach to bootstrapping instead of Monte Carlo rollouts (in the form of a product of two value estimates), which tend to produce lower variance estimates. We will emphasize this in the introduction and related work.
>
> > Q3. Section 5.3 states that a solution to the COE equation is $0$ . However, on page 6 the text says that values cannot decrease. Given that values can easily be initialized to be greater than $0$.
>
> Values cannot decrease for the tabular case, where we take maximum over all possible subgoals. Section 5.3 addresses generalization to the Deep-RL scenario, where we use a subgoal generator instead of having maximum over subgoals. Therefore, the value can decrease. Moreover, note that the reason why tabular COE-VI cannot decrease is the initialization of transitions to be 1 and $\gamma$ (lines 2-4) of Algorithm 1. This initialization assures that the algorithm can always choose at least the Bellman or the “null” decomposition, which keeps the value constant. Grounding update imitates this initialization for deep RL by updating the values of transitioning states to be $\gamma$.
>
> > Q4. It also seems that since the update equation is multiplicative and all values are initialized to some value $v \leq 1$, a value can never be larger than $1$. However, if I imagine an MDP with a single state absorbing state that is also the goal, the reward in that state would be $\gamma$ by definition. The maximum value I can obtain can clearly be larger than $1$. Can you explain where my logic is flawed?
>
> We are missing the assumption that every goal state is also a terminal state. We thank the reviewer for pointing it out.
>
> > Q5. In fact, unless I am missing something, values will always approach $\gamma$
>  because the largest value product will always be $1 \star \gamma$. Why is this not the case?
>
> As Above
>
> > Q6. In section 5.4, why would Q(s, a, s’) always be equal to $\gamma$? Same equation, I am not sure what “$Q_{Bellman}$” means? The Q-function should be defined according to section 3.
>
> 6. This is because we assume in section 5.4 that the Grounding Update is well-fitted, so that $Q(s, a, s')$ is close to $\gamma$: $(Q(s, a, s’) \approx \gamma$. There was a typo here. We thank the reviewer for pointing it out. In practice, Q(s, a, s’) will not be exactly $\gamma$, so this equation is approximate. We changed the equality to $\approx$ in the new revision for clarity.
>
> > Q7. If value iteration is a special case of the algorithm you propose, how can your algorithm be significantly faster?
>
> 7. Value Iteration is a special case of Generalized Compositional Value Iteration with the function ProduceSubgoal producing the next transition state. However, COE-Value Iteration is also a special case of Generalized Compositional Value iteration, so they don’t share this relation. We will make the statements more clear.
>
> > Q8. In the second part of the proof, it states something like “From our assumption $V_{k-1}(s, subg) = V^*(s, subg), V_{k-1}(subg, g) = V^*(subg, g).$" Why can we assume that these are optimal?
>
> This is a part of the inductive assumption for the $k$th step.
>
> > Q9. The update equation in line 9 of the algorithm is missing what is being updated.
>
> It’s true, $V_2(s, g)$ is being updated. We thank the reviewer for pointing it out.
>
> For the next revision, we also corrected the typos and unclear definitions mentioned by the reviewer.
>
> [1] - Corinna Cortes, Yishay Mansour, and Mehryar Mohri. Learning bounds for importance weighting
>
> [2] - Alberto Maria Metelli, Matteo Papini, Francesco Faccio, and Marcello Restelli. Policy optimization
> via importance sampling
>
> [3] - Anna Harutyunyan, Marc G Bellemare, Tom Stepleton, and Remi Munos. Q (λ) with off-policy
> corrections.

---

> > ### Comment · Reviewer_EvmW · 2023-11-21
> >
> > Thank you for the clarifications. I went ahead and read the other reviews and responses as well. As I understand, the authors are withdrawing the submissions but I figured feedback on this response might be useful for future iterations.
> >
> > > Every episode has a single goal
> >
> > Your current MDP framework does not consider anything that would commonly be referred to as an episode and neither does your algorithm.
> >
> > > However, they rely on Monte-Carlo rollouts, and as a consequence
> >
> > There seems to be consistent confusion about deep RL approaches and theoretical approaches but the manuscript tries to do both. It might be good to have a clearer differentiation in the text that will facilitate the reader to make this distinction.
> >
> > > We are missing the assumption that every goal state is also a terminal state.
> >
> > There is also no such thing as a terminal state in the MDPs that are defined in the manuscript and it is not clear what this would mean.
> >
> > > Re Q6
> >
> > The confusion in Q6 in part arises from the fact that the $Q$ function all of a sudden depends on a next state which is not defined.
> >
> > > Re Q7
> >
> > In general, the number of iterations of an algorithm is irrelevant for computational purposes if the number of accesses in the model of computation is too large. I find it counter-intuitive to use a model of computation that is not defined or explained and differs from the commonly applied one. It might make sense in a future submission to use the standard model of computation.
> >
> > > Re Q8
> >
> > The base case for this induction would have to be something like $V_0 (s, subg) = V^*(s, subg)$ if I understand correctly, which seems like it cannot be true and is also not stated in the manuscript.
> >
> > I hope these suggestions are helpful for future submissions. For this manuscript, I will retain my score.

---

### Official Review · Reviewer_v7VZ · 2023-10-31

**Soundness:** 1 poor
**Presentation:** 1 poor
**Contribution:** 1 poor
**Rating:** 3
**Confidence:** 4

**Summary:**

The paper concerns Goal-conditioned RL. The authors propose Compositional Value Interaction, an adaptation of the standard Value Iteration that takes into account sub-goals, i.e., the value function is defined in terms of states and sub-goals. The authors then provide some theoretical results demonstrating the convergence of the new value iteration. Experiments are conducted based on gridworld environments to compare Compositional Value Iterations with standard RL methods.

## after rebuttal:

Some issues still remain. For instance, the deterministic setting and the limitation of the current experiments, which are only based on two simple gridworlds.

I've increased my score. I think the paper is still not ready for publication.

**Strengths:**

The paper addresses an important problem.

**Weaknesses:**

The writing and exposition are very poor. The paper is unreadable. I got lost at the very beginning of the paper and cannot understand the formulations.
   -  On Page 3: The definitions of $v_{\pi}(s)$ and $G_t$ do not look reasonable. In the definition of $v_{\pi}(s)$, why does $t$ only appear on the right-hand side? Moreover, the expectation is conditional on $S_t = s$, but then when $G_t$ is defined, the expectation is, again, conditional on the same thing, $S_t = s$. The "Bellman Expectation equation" does not look right as well - what is $S_t + 1$"?
   - On Page 4, $V^{\pi}(s,g)$ is never defined. Previously, $V(.)$ was just defined as a function of only $s$. Because of this, I totally got lost when encountering formulations involving $V(s,g)$. For instance, the equation above Lemma 1 is nonsense to me.
  - Many "full stop" dots appear at the beginnings of sentences.
  - First line of Page 5: $d_{\pi}(s,g)$ is defined as the distance between $s$ and $g$, but in terms of what? Why is $d_{\pi}(s,g) = |T_{\pi}(s,g)|-1$? .
 - In the fourth line of Page 5, why is $V^{\pi}(s,g) = E[\gamma^{d_{\pi}(s,g)}]$? Where does this come from? Since I got lost, I do not
understand the subsequent results, so I'm not able to validate them.

Besides the poor exposition, the deterministic setting is also very limited.

The experiments are conducted based on two simple gridworld environments, which is also very limited. More advanced environments should be used, such as Mujoco or OpenAI tasks. Moreover, the authors only compared their approach with two baselines. This is far from being sufficient.

**Questions:**

Please see the Weaknesses.

---

> ### Author Response · Authors · 2023-11-19
>
> We thank the reviewer for their valuable feedback and suggestions.
> We are somewhat surprised by the reviewer's comments about the paper being "unreadable," especially since other reviewers (Reviewer jWih and Reviewer EvmW) consider the writing and structure of the paper to be one of its strengths.
> It is possible that typos and slight inconsistencies in the formal definitions might have caused difficulties.
> We have fixed them in the new revision and kindly invite the reviewer to read our revised submission.
>
> Regarding the assumption about deterministic settings, as mentioned in our general response, the deterministic setting is primarily a consideration for our theoretical analysis.
> We believe that our approach could provide sample-efficiency gains even in stochastic environments.
> Our experiments in the continuous gridworld show that our approach can handle noisy transitions in practice.
>
> Answers to questions:
>
> We thank the reviewer for pointing out the typos in the background. We have fixed them in the revised version.
>
> >On Page 4, $V^\pi(s, g)$ is never defined. Previously, $V(.)$ was just defined as a function of only $s$. Because of this, I totally got lost when encountering formulations involving $V(s, g)$. For instance, the equation above Lemma 1 is nonsense to me.
>
> $V(s, g)$ is defined implicitly by the last paragraph of the background, which states that in goal-conditioned RL $\mathcal{S} = \mathcal{S}_s \times \mathcal{S}_g$ and since $V: \mathcal{S} \rightarrow \mathbf{R}$ it directly becomes $V: \mathcal{S}_s \times \mathcal{S}_g \rightarrow \mathbf{R}$. This, however, was causing a bit of confusion also in other reviews, so we have clarified this in our updated version.
>
> > First line of Page 5: $d_\pi(s,g)$ is defined as the distance between $s$ and $g$, but in terms of what? Why is $d_\pi(s, g)=|T_\pi(s,g)| - 1$?
>
> $d_\pi(s, g)=|T_\pi(s,g)| - 1$, because that's the formal definition of $d_\pi(s, g)$. The note about the distance is to make the definition more accessible. This quantity represents the number of steps policy $\pi$ has to execute to reach $g$ from $s$. We have clarified this bit in the new revision.
>
> > In the fourth line of Page 5, why is $V^\pi(s, g) = E[\gamma^{d_pi(s,g)}]$? Where does this come from? Since I got lost, I do not understand the subsequent results, so I'm not able to validate them
>
> We thank the reviewer for the comment. We have attempted to clarify this in the new revision. To reiterate it here, this is a direct application of the definition of value function combined with the assumption about the reward from the environment and the definition of $d_\pi(s,g)$.
>
> We hope that our response and updated version clarify the aspects needed to evaluate our work.

---

> ### Comment · Reviewer_v7VZ · 2023-11-22
> **Responses**
>
> I thank the authors for the clarifications. I also noticed some updates in the paper, which make the paper easier to read.
>
> Some issues still remain. For instance, the deterministic setting and the limitation of the current experiments, which are only based on two simple gridworlds.
>
> I increased my score. I think the paper is still not ready for publication.

---

### Author Response · Authors · 2023-11-19

We would like to thank all reviewers for their thoughtful comments and valuable feedback.

The feedback has helped us better understand how the paper can be improved.
We also appreciate the reviewers pointing out inconsistencies in our notation.

After thoughtful consideration, we have decided that we will withdraw the paper and rewrite it for the next submission. The main reason for the withdrawal was the similarity of the tabular version of COE to Floyd-Warshall Reinforcement Learning (FWRL) pointed out by the reviewer jWih. We were not aware of FWRL before the submission. We consider the generalization to deep RL our main contribution and view the tabular derivations as a motivation for that. However, clearly explaining our contribution and putting it in context to FWRL requires the paper to be rewritten.

Out of courtesy to the reviewers, we still decided to answer the reviewers' questions. We also provide an updated version of the paper addressing the remaining points raised by the reviewers.

One common concern in all the reviews is about the assumption of a deterministic environment. While we agree that this is indeed a limitation,
this is mainly a simplifying assumption that is necessary for our theoretical analysis. We aim to extend our theoretical analysis to the more general stochastic setting in the future. The fundamental difficulty in moving beyond the deterministic setting is that one can create scenarios where it is impossible to determine the optimal subgoal before the stochastic transition occurs.

However, we believe that our approach can provide sample efficiency gains in environments with stochastic transitions.
Note that our experimental evaluation on the continuous gridworld involves stochastic transitions in the form of added noise at each move.

We respond to individual points raised by reviewers in separate replies to each review below.